# Observations on the Malting of Ancient Wheats: Einkorn, Emmer and Spelt

**Alice Fujita, Senay Simsek**  **and Paul B. Schwarz \***

Department of Plant Sciences, North Dakota State University, Fargo, ND 58108, USA;
fujitaalice@yahoo.com (A.F.); senay.simsek@ndsu.edu (S.S.)
**\*** Correspondence: paul.schwarz@ndsu.edu; Tel.: +1-701-231-7732

**Abstract:** There have been tremendous marketing efforts and consumer interest in the so-called ancient grains. Einkorn, emmer and spelt, which are sometimes referred to as ancient wheats, are frequently included in this category, and have gained some attention among brewers. The objective of the current study was to compare the malting behavior and quality of einkorn, emmer and spelt cultivars obtained from the same growing environment. Aside from standard malt quality traits, the levels of β-amylase, protease, xylanase, wort arabinoxylans and wort phenolic acids were measured. While protein levels of the samples were higher (11.4–14.0%) than normally selected for wheat malt, the results indicated that malts of acceptable quality in terms of extract and amylolytic activity can be prepared from the three grain types. However, the ideal malting protocol will likely differ between the grains. The kernels of einkorn are significantly smaller, and steep hydration and malt modification are quicker. In terms of potential health benefits from antioxidant capacity and dietary fiber, wort from einkorn trended to higher levels of free and conjugated ferulic acid, as well as high-molecular-weight arabinoxylan.

**Keywords:** arabinoxylan; brewing; einkorn; enzyme activity; emmer; malt; phenolic acid; spelt; and sprouting

## 1. Introduction

There is a growing awareness of less-utilized types of grains/seeds, which are often marketed as healthier alternatives to commodity crops and are sometimes referred to as ancient grains or superfoods. Consumers are looking for alternatives to incorporate more fiber, whole grains, and plant-based protein into their diets [1], and in response, food companies began extensively marketing ancient grains over the past five to ten years. In 2018, the global market for ancient grains was estimated at 457 million US dollars and is expected to achieve six billion dollars by 2024 [2].

The so-called ancient grains encompass an array of cereals and pseudo cereals, and while there is no official definition, they are often described as being largely unimproved through plant breeding efforts [3], and by default, some individuals presume them to be healthier. While this is not strictly true, these crops have generally received less attention than bread wheat, rice, maize and barley. In terms of the ancient wheats, einkorn, emmer, spelt, and the trademarked Kamut are most often considered. These grains are all commercially available as whole grain, flour, and as ingredients in processed products. The most common and commercially available ancient wheats are einkorn (*Triticum monococcum* var. *Monococcum*. Diploid genome AA), emmer (*Triticum turgidum* var. *Dicoccum*. Tetraploid genome AABB) and spelt (*Triticum aestivum* var. *Spelta*. Hexaploid genome AABBDD) [4,5]. These grains are, in general, considered hulled wheats [6], which means they are not free threshing at harvest, and additional processing is necessary for removal of the hull [7]. Einkorn was first domesticated in what is today Turkey in approximately 11,000 BCE, while cultivation of emmer started

in approximately 10,500 BCE in the "Fertile Crescent" [8]. For spelt, the most accepted origin is the region from Transcaucasia to southwestern Caspian Iran [6] in approximately 6000 BCE [9]. Today, these crops are often cultivated in organic farm operations under low-input systems [10]. This type of cropping has been reported to be beneficial for the production of bioactive compounds, such as phenolics acids and flavonoids, as these compounds are secondary metabolites produced by the plants under stress [11]. Studies have shown that old wheat varieties had higher concentrations of phenolics and important minerals than modern varieties [12]. The antioxidant capacity of phenolic compounds is well known, and consequently they may promote health and prevent or reduce the risk of chronic diseases, such as cancer, cardiovascular disease, hypertension, type 2 diabetes, and obesity [13–16].

The brewing industry is not really different from the overall food industry in that many consumers are also looking for unique characteristics, new ingredients, and local flavor [17]. While barley malt-based lagers and ales account for the vast majority of the global beer market, wheat beers are an important segment, and are produced by many craft brewers, as well as by several large multi-national brewers. Bond et al. [18] estimated that wheat malt accounted for approximately 5–10% of all malt use in the USA. However, this malt is likely always produced from hexaploid common wheats (*Triticum aestivum* var. *Aestivum* AABBDD genome). Nevertheless, a search of the global consumer website Ratebeer.com (https://www.ratebeer.com) [19] showed beers being produced, in part, from either einkorn, emmer or spelt. These are generally from smaller breweries in Europe or North America. Several recent publications have cited potential health benefits of these grains in brewing [8,12,20–22]. At the time of publication, malted spelt was available from several German maltsters including BestMalz, (Heidelberg, https://bestmalz.de/) [23] and Wyermann (Bamberg, https://www.weyermann.de/) [24]. Spelt, einkorn and emmer malts are also likely produced on a limited scale by craft maltsters, and an important secondary market for some is sprouted grains for the food industry.

Interest in the use of these grains in brewing may also, to a limited degree, derive from historical considerations. The origins of brewing with cereal grains, of course, can be traced back millennia, to or near, the same regions where domestication occurred. The probable use of emmer, barley, and einkorn for brewing at several ancient Near Eastern sites has been documented [25–27]. Evidence suggests that both malted and raw grains were used. Cereals reached Europe between 8000 and 6000 years ago [28], and references to early brewing use of these grains can be found in the publication by Nelson [29]. Brewing from the iron age through medieval times was entirely dependent upon locally available grains [30]. Hornsey's History of Beer and Brewing [31] makes reference to Jacobus Theodorus's 1588 Neuw Kreuterbuch (herbal book or botanical), which begins to describe the malting and brewing processes as 'They take wheat, barley, spelt, rye or oats, either one kind (for good beer can be prepared from all these cereals) or two together; they steep...', which further reinforces the historical importance of multiple local grains in beer production. Despite the use of ancient wheats in European brewing we are not aware of any style that can be associated specifically with any of these wheat grains. Nevertheless, it must be considered that today's taxonomic designations are relatively recent, and the differentiation between wheat types is not necessarily the same in the historical literature. While multiple types of wheat were once used, the cultivation of hexaploid common wheats likely had replaced the ancient wheats in many regions by the advent of modern/industrial brewing in the early to mid-19th century.

Recent years have seen increased research interest on ancient wheats, and in terms of germinated grains, it has either been from the perspective of sprouting for use in breads and other foods [16,32–34], or from interest in brewing [12,20,35,36]. It is well known that the malting of wheat presents challenges when compared to barley [37], and it would be anticipated that the ancient wheats would present similar issues, particularly if dehulled prior to malting. Lack of the hull (husk) results in more dense packing of grain in the steep and can also result in damage to the growing acrospire, and lautering problems. The dehulling of ancient wheats can result in mechanical damage and reduced germination, which has been reported in the case of research on the malting of einkorn [10]. Researchers at the Italian Brewing Research Centre compared the malting and brewing properties of three hulled and



dehulled emmer samples, and a single einkorn cultivar [35,38]. The hulled samples needed a longer time for germination and the presence of hulls also resulted in lower extract yields compared to dehulled samples. However, beer prepared from hulled emmer malt had good sensory properties [37]. Sachambula et al. [21], evaluated the influence of steep moisture and germination times on the quality of einkorn malt. Results suggested that 45% moisture with 4 days of germination were the best conditions for malt quality. Muñoz-Insa et al. [36] utilized response surface methodology to study the impact of steep moisture, germination time and germination temperature on spelt malt quality. Optimal malt quality was achieved with 47% moisture, and 5 days of germination at 17 °C.

The aim of this study was to evaluate the malting behavior and quality of einkorn, emmer and spelt from the same growing environment. Durum and hard red spring wheat cultivars were included, for respective comparison to emmer and spelt. Aside from basic grain and malt quality parameters, water uptake and hydration, enzyme development, total arabinoxylan, high-molecular-weight arabinoxylan and phenolic acids in wort were also evaluated. Results were compared with those of barley and commercial wheat malt.

## 2. Materials and Methods

### 2.1. Materials

Samples of einkorn (N = 3 varieties), emmer (N = 4) and spelt (N = 3) were from organic variety trials at the NDSU Carrington Research Extension Center in 2018 (Table 1). Durum wheat (N = 5) and hard red spring wheat (N = 5) from the same trial were included for comparative purposes. The hulled wheats were dehulled using an impact dehuller described by Doehlert and McMullen [39].

As brewers are more familiar with barley malt quality, two barley cultivars were malted in conjunction with the wheat samples. These included CDC Meredith and KWS Fantex. Both were from the 2018 crop and were grown at University test plots in North Dakota and Maine, respectively. CDC Meredith was developed by the Crop Development Centre in Saskatoon, Canada, and is an adjunct brewing type two-rowed barley. KWS Fantex is from KWS in Germany and represents more of a European all-malt quality profile. Commercial malts were also included as controls. These included two samples (A and B) of 2-rowed pale malts from two different USA maltsters, and one sample of white wheat malt from an USA maltster. As these were purchased from a secondary brewer's supply source, neither the cultivar or crop year are known.

### 2.2. Grain Quality Tests

One thousand kernel weight was determined following Barley Method-2D of the American Society of Brewing Chemists (ASBC) [40]. Protein percentage was measured by nitrogen combustion (AACC-1 approved method 46-30.01) [41] using a LECO FP528 (LECO, St. Joseph, MI, USA) nitrogen/protein analyzer. Falling number were measured according to AACC International Method 56-81.03 at Falling Number 1900 (Perten instruments, Springfield, IL, USA). Deoxynivalenol (DON) was measured by GC-ECD [41].

Germinative energy was determined according to ASBC Barley Method-3A [40]. Hydration Index (Chapon test) were determined as described by Turner et al. [42]. Steeping time to reach 45% moisture was determined on a 10 g (dry basis) sample according to Banasik et al. [43]. Malting loss was calculated as the difference in weight (dry basis) between the ungerminated grain and the clean malt [44].

**Table 1.** Grain quality of wheat samples and barley controls screened for micro-malting.

| Grain | Cultivar | 1000 Kernel Weight (g) | | Protein (%) | | Falling Number (s) | | Grain DON (µg/g) | | Malt DON(µg/g) | |
|---|---|---|---|---|---|---|---|---|---|---|---|
| | | Value | Average | Value | Average | Value | Average | Value | Average | Value | Average |
| Einkorn | TM23 | 30.2 | | 14.0 | | 385 | | 0.03 | | 1.0 | |
| | WB Alpine | 29.3 | 29.5 ± 0.3 c | 12.7 | 13.3 ± 0.3 bc | 393 | 383.3 ± 6.1 bc | 0.03 | 0.03 ± 0.0 a | 0.2 | 0.6 ± 0.3 ab |
| | PI 538722 | 29.0 | | 13.2 | | 372 | | 0.03 | | ne | |
| Emmer | Vernal | 34.8 | | 12.5 | | 467 | | 0.10 | | ne | |
| | Lucile | 35.3 | | 12.3 | | 485 | | 0.05 | | 2.2 | |
| | ND Common | 32.8 | 34.0 ± 0.6 b | 11.7 | 12.0 ± 0.25 d | 464 | 475.8 ± 5.9 a | 0.0 | 0.04 ± 0.03 a | 1.3 | 1.2 ± 0.5 a |
| | Yaroslav | 33.2 | | 11.4 | | 487 | | 0.0 | | 0.1 | |
| Spelt | CDC Zorba | 36.0 | | 12.1 | | 270 | | 0.0 | | 0.0 | |
| | 94-288 | 37.1 | 38.6 ± 2.1 a | 12.4 | 12.5 ± 0.3 cd | 445 | 363.0 ± 50.8 c | 0.0 | 0.02 ± 0.02 a | 0.0 | 0.0 ± 0.0 b |
| | SK3P | 42.7 | | 13.0 | | 374 | | 0.05 | | 0.0 | |
| Durum | Mountrail | ne | | ne | | 402 | | 0.08 | | ne | |
| | ND Riverland | 38.5 | | 14.2 | | 421 | | 0.05 | | 1.3 | |
| | Joppa | 37.0 | 38.6 ± 0.76 a | 13.8 | 14.1 ± 0.1 b | 429 | 426.4 ± 8.2 ab | 0.0 | 0.09 ± 0.05 a | 1.1 | 1.3 ± 0.07 a |
| | Divide | 40.4 | | 14.2 | | 453 | | 0.0 | | 1.4 | |
| | VTPeak | ne | | ne | | 427 | | 0.3 | | ne | |
| HRS | Barlow | 34.2 | | 15.3 | | 387 | | 0.0 | | 0.0 | |
| | Glenn | 34.8 | | 16.4 | | 379 | | 0.0 | | 0.1 | |
| | Linkert | 37.9 | 35.3 ± 0.7 b | 16.1 | 16.0 ± 0.4 a | 434 | 416.8 ± 18.8 bc | 0.0 | 0.0 ± 0.0 a | 0.1 | 0.08 ± 0.02 b |
| | SY Ingmar | 34.0 | | 17.3 | | 482 | | 0.0 | | 0.1 | |
| | TCG Spitfire | 35.4 | | 14.8 | | 402 | | 0.0 | | ne | |
| Barley | CDC Meredith | 43.6 | nc | 14.3 | nc | ne | nc | 0.0 | nc | 0.0 | nc |
| | KWS Fantex | 43.9 | | 9.4 | | ne | | 0.0 | | 0.0 | |

Notes: DON—deoxynivalenol; ne—not evaluated; nc—not calculated, barley was not included in the statistical determinations; different letters on the average values within columns signify significant differences ($p < 0.05$).

### 2.3. Micro-Malting

One sample (80 dry basis) of each of the selected samples were micro-malted according to our standard method [45]. Samples were steeped to achieve 45% moisture. Steeping involved alternating 11 h of aerated immersions followed by 1 h air rests. Germination was performed at 16 °C, and 95% relative humidity. Distilled water was added to the samples daily to account for moisture loss. After 4 days of germination, the samples were kilned. Kilning temperatures were ramped from 49 to 85 °C over 24 h. After drying, the samples were de-rooted and stored at 4 °C. For studies involving the time-course development of enzyme activity during malting, 20 g samples were malted according to the above protocol. A separate (20 g) sample was used for each time point, and steeped and green malt samples were freeze-dried.

### 2.4. Determination of Malt Quality

Malt moisture, malt extract, wort soluble protein, free amino nitrogen (FAN), wort color, α-amylase, diastatic power (DP), β-glucan and viscosity were assessed according to ASBC official [40] or slightly modified methods as previously described [46]. Because of the lack of husk, wheat malt samples were centrifuged ($3000\times g$, 20 °C, 10 min) prior to filtration for determination of malt extract. Wort turbidity was determined with a Hach 2100N Laboratory Turbidimeter (Hach Company, Loveland, CO, USA).

### 2.5. Determination of α- and β-Amylase, Protease and Xylanase Activities in Germinating Grain and Finished Malt

All additional enzyme activities were determined using test kits or reagents from Megazyme International Ireland Ltd. (Wicklow, Ireland). Alpha-amylase and β-amylase activities were determined using the Malt-Amylase Assay procedure (Ceralpha and betamyl-3 respectively). One unit of activity was defined as the amount of enzyme to release 1 μmol of ρ-nitrophenol per minute. Endo-protease activity was evaluated using azurine-crosslinked casein substrate (Protazyme AK tablets). One unit of enzyme activity was defined as the change in the absorbance per hour per gram of sample. Endo-β-(1,4)-ᴅ-xylanase activities were analyzed using an azo-wheat arabinoxylan kit were expressed in mU/g db. Samples of sound grain, kilned malt, and germinating samples at steep out and 1–4 days of germinated were evaluated for all activities. Steep-out and germinated samples were lyophilized prior to analysis.

#### 2.5.1. Determination of Wort Arabinoxylans (AX)

Total arabinoxylans and AX ratio were measured according to Blakeney [47]. Further, high-molecular-weight AX was determined precipitating the wort with cool 80% ethanol (*v/v*) overnight at 4 °C, then centrifuged at 10,000 rpm/30 min at 8 °C. The precipitated was freeze-dried and measured the same way as total arabinoxylans. Arabinose and xylose were used as monosaccharide standards in the analysis. Myo-inositol was used as the internal standard. After derivatization of alditol acetate samples were analyzed using Hewlet Packard 5890 series II GC system with a flame ionization detector (FID) (Agilent Technologies, Inc. Santa Clara, CA, USA). Separation was performed with Supelco SP-2380 fused silica capillary column (30m × 0.25 mm × 2 μm) (Supelco Bellefonte, PA, USA) [41]. The system parameters were as follows: flow rate, 0.8 mL/min; flow pressure, 82,737 Pa; oven temperature, 100 °C; detector temperature, 250 °C; and injector temperature, 230 °C. The carrier gas was helium. Arabinoxylan content in acetone was calculated as 0.88 × (Arabinose + Xylose).

#### 2.5.2. Determination of Wort Phenolic Acids

Phenolic acids were analyzed following the methods of Kim [48] with some minor modifications described by Wang [46]. Two forms, free and soluble conjugated were extracted from wort samples, then, analyzed on an Agilent 1290 series liquid chromatography with a 6540 UHD Accurate-Mass Quadrupole Time-of-flight (LC/MS-Q-TOF, Agilent Technologies, Santa Clara, CA, USA). Separation was performed

on Zorbax SB-C18 column (1.8 μm, 2.1 mm × 50 mm, Agilent) at 30 °C. The mobile phase used was 0.1% of formic acid in acetonitrile (solvent A) and 0.1% of formic acid in water (solvent B). The flow rate was 0.4 mL/min for 15 min run. Gradient conditions were: 0–1 min isocratic with 0.3% B; then the linear increase from 3 to 97% B for 1–10 min; shifting back to an initial setting for 5 min. Post run of 4 min. The AJS electrospray ionization interference (ESI) was used in the positive mode.

*2.6. Statistical Analysis*

Analyses of variance (ANOVAs) were performed using JMP Pro (version 15, SAS. Cary, NC, USA) for least significant difference (LSD) at $p < 0.05$ comparing different types of grain. Further, Pearson's correlation coefficients were analyzed for malt extract, sound sample's protein contents, DP, α-amylase, soluble protein, S/T, FAN, viscosity, color, β-glucan, Megazyme kits (protease, xylanase, α-amylase, β-amylase), AX HMW, ferulic acids, and total phenolic acids contents of the worts samples. Finally, hierarchical cluster analysis (HCA) for all results was analyzed to check the similarity.

## 3. Results

*3.1. Grain Quality*

The hulled wheats were dehulled using an impact dehuller as part of another study [49]. Some damage was noted with emmer and spelt, but dehulling resulted in a large percentage of broken kernels in the einkorn samples (Figure 1). For purposes of this study, broken kernels were removed by hand. However, this is obviously not practical on a larger scale.

The results of grain quality analysis are shown in Table 1, and germination results in Table 2. One thousand kernel weight (1000 kw) is an indicator of kernel size. Significant differences were seen between grain types when averaged across cultivars, with the exception of HRS-emmer, and spelt-durum. Einkorn had the lowest 1000 kw (29.5 g), followed by emmer (34.0 g) and then HRS wheat (35.3 g). Durum and spelt (both 38.6 g) had the highest 1000 kw. Photographs of the various wheat types are shown in Figure 1 for visual comparison. Einkorn kernels tend to be flatter or narrower in width than the other wheat types.

Although not measured in the current study, data on hectoliter weight (kg/hL or bushel weight in lbs/BU) may also have been instructive. While there is a significant relationship between 1000 kw and hectoliter weight in wheat, it is not strong [50]. The higher test weight of common wheat (77.2 kg/hL, 60 lb/BU) vs. barley (61.8 kg/hL, 48 lbs/BU) results in more dense packing in the steep, and swelling upon hydration can make it more difficult to handle. The standard weight of dehulled spelt is similar to that of common wheat [51], and it would be expected that all dehulled wheat types would present similar challenges. The small grain size and flatter kernels of einkorn could also pose a problem to some maltsters in terms of retention on malting floors or in drums, depending on the width of openings.

Among the ancient wheat types in this study, einkorn trended towards higher protein contents (13.3%), and emmer towards lower levels (12.0%). However, these differences were not large, and variability was seen between varieties within wheat types. Average protein content of the HRS samples was the highest (16.0%), and the average for durum (14.1%) was higher than that of the ancient wheats. However, the number of samples used in the current study was quite limited and represented only a single growing environment.

In addition to protein and 1000 kw, several factors that can be detrimental to malting were also measured. Falling number is a measure of preharvest sprouting (PHS), and values of >350 s are generally believed to indicate sound wheat [52]. PHS is an issue in malting as severely sprouted samples may show lower germination, or germination may decrease with grain storage time. With the exception of one variety of spelt (CDC Zorba, 270 s), the samples appeared to be free from PHS. However, as this sample still showed good germinative energy (98%, Table 2), it was included in further experiments.

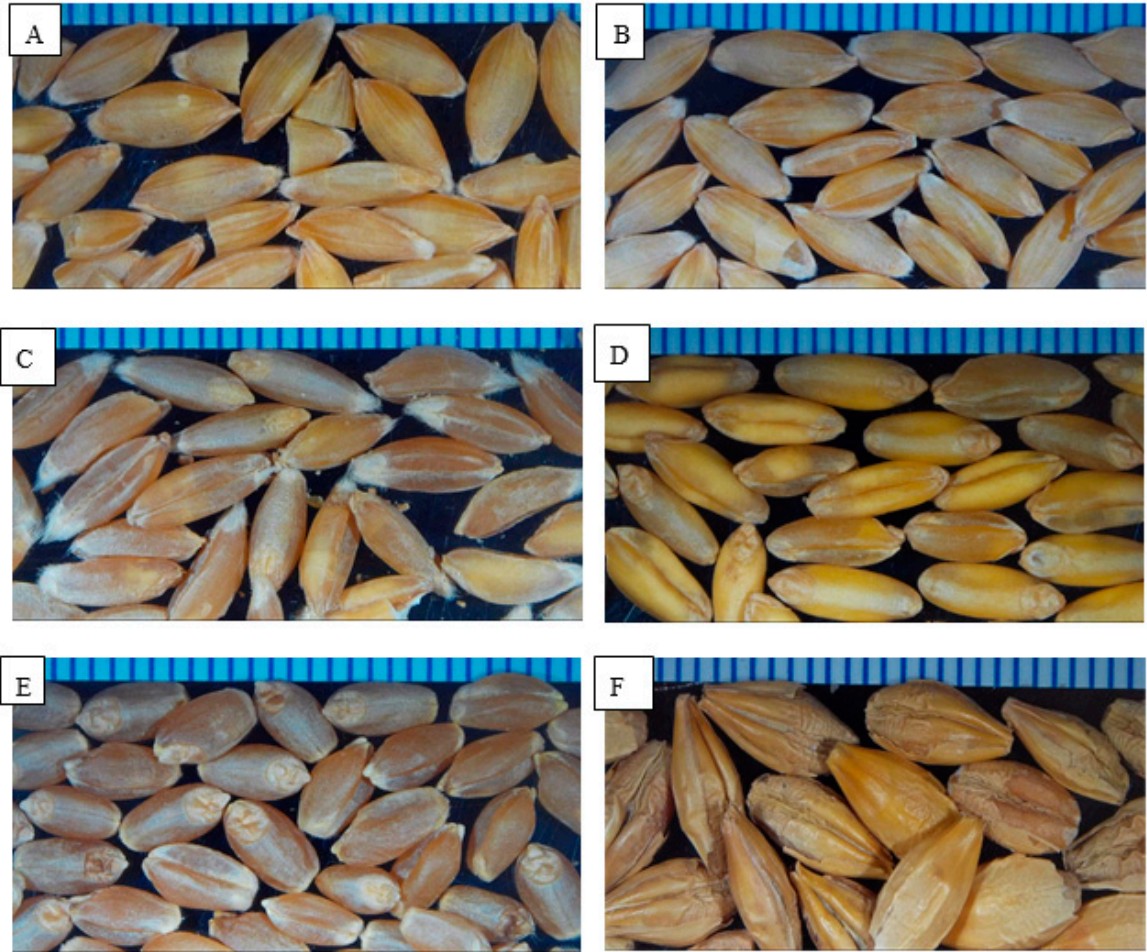

**Figure 1.** Photographs of grain samples: (**A**) einkorn (WB Alpine), (**B**) emmer (ND Common), (**C**) spelt (CDC Zorba), (**D**), durum (Joppa), (**E**) HRS wheat (Glenn), and (**F**) barley (KWS Fantex). Scale is in mm.

As malting is a process of controlled germination, a high percentage of germinative energy is required, which in the case of barley, is typically specified as ≥95%. The requirements for wheat should be similar, but considering potential mechanical damage in dehulling, the specifications for hulled wheats might need to be set slightly lower. As shown in Table 2, all samples were ≥93%, with the exception of four durum wheat varieties. Mountrail and VT Peak were excluded from further experiments because of very poor germination. Although there was some variation seen within wheat type, overall germination was highest in HRS wheat and spelt, and poorest in durum.

Finally, levels of the Fusarium mycotoxin, deoxynivalenol (DON) were measured in grain and malt samples. This is as the occurrence of Fusarium Head Blight (FHB) in common in many cereal production regions, and the levels of DON have been shown to significantly increase during the malting of FHB infected wheat [53]. While many samples were free of DON, low levels were found on all einkorn samples, and several of the emmer, spelt and durum samples. However, following malting, DON increased to ≥ 1.0µg/g only in einkorn TM23, emmer varieties, Lucile and ND Common, and durum varieties, ND Riverland, Joppa, and Divide. This is of concern as Fusarium infection can impact the results of malt analysis, particularly in terms of soluble protein. However, a study on barley had shown that these impacts on quality were not pronounced when DON levels were ≤1.0 µg/g [54]. No samples were eliminated from further study based upon DON levels, but Fusarium infection should be considered in the interpretation of malt data for these samples, particularly if malt results seem atypical.

**Table 2.** Germinative energy, hydration index, steeping time and malt loss for wheat samples and barley controls.

| Grain | Cultivar | Germinative Energy (%) | | | Hydration Index (%) | | | Steeping Time (h to 45% Moisture) | Malt Loss (%) |
|---|---|---|---|---|---|---|---|---|---|
| | | 24 h | 48 h | 72 h | 24 h | 48 h | 72 h | | |
| Einkorn | TM23 | 89.3 a | 93.0 bc | 93.0 bc | 76 a | 97 a | 100 a | 56.2 b | 10.6 a |
| | WB Alpine | 93.8 a | 98.8 bc | 98.8 bc | 63 a | 100 a | 100 a | 42.5 b | 10.7 a |
| Emmer | Vernal | 89.3 a | 92.3 c | 92.8 cd | 38 b | 81 b | 97 a | 74.7 a | ne |
| | Lucile | 87.8 a | 92.3 c | 92.5 cd | 34 b | 69 b | 95 a | 65.7 a | 11.1 a |
| | ND Common | 89.8 a | 94.3 c | 94.3 cd | 44 b | 75 b | 99 a | 67.5 a | 12.8 a |
| | Yaroslav | 82.5 a | 93.0 c | 93.3 cd | 37 b | 79 b | 99 a | 55.0 a | 10.9 a |
| Spelt | CDC Zorba | 88.7 a | 98.0 ab | 97.7 ab | 39 bc | 78 b | 94 a | 47.6 ab | 10.4 b |
| | 94-288 | 94.8 a | 98.0 ab | 98.5 ab | 33 bc | 71 b | 94 a | 60.2 ab | 7.8 b |
| | SK3P | 82.5 a | 97.3 ab | 98.3 ab | 25 bc | 62 b | 90 a | 61.1 ab | 7.5 b |
| Durum | Mountrail | 55.8 b | 86.0 d | 88.3 d | ne | ne | ne | ne | ne |
| | ND Riverland | 42.8 b | 89.8 d | 92.0 d | 35 c | 71 bc | 98 ab | 67.0 a | 9.1 b |
| | Joppa | 45.8 b | 90.8 d | 93.0 d | 23 c | 53 bc | 86 ab | 63.8 a | 8.3 b |
| | Divide | 60.8 b | 89.5 d | 91.0 d | 16 c | 53 bc | 84 ab | 56.8 a | 8.1 b |
| | VTPeak | 57.0 b | 89.3 d | 90.0 d | ne | ne | ne | ne | ne |
| HRS | Barlow | 89.7 a | 99.7 a | 99.7 a | 13 d | 47 c | 88 b | 63.6 a | 8.1 b |
| | Glenn | 90.7 a | 99.3 a | 99.3 a | 10 d | 47 c | 81 b | 65.5 a | 8.5 b |
| | Linkert | 71.7 a | 99.0 a | 99.3 a | 13 d | 55 c | 84 b | 64.7 a | 9.2 b |
| | SY Ingmar | 91.3 a | 99.7 a | 99.7 a | 3 d | 27 c | 61 b | 71.7 a | 8.8 b |
| Barley | CDC Meredith | ne | ne | ne | 20 | 61 | 90 | 43.9 | 7.9 |
| | KWS Fantex | ne | ne | ne | 42 | 87 | 100 | 27.1 | 9.2 |

Note: ne—not evaluated; different letters on the average values within columns signify significant differences ($p < 0.05$). Barley was not included in the statistical determinations.

### 3.2. Water Uptake and Hydration

The proper steeping of grain is essential in ensuring proper endosperm modification and good malt quality. With barley, it is known to be influenced by grain size, protein content and endosperm hardness/vitreousness [37]. In the current study both water uptake rate (steep time), and endosperm hydration index were considered. The time required for each sample to reach 45% moisture content is shown in Table 2. The steep times for all wheat types were longer than those of the barley control samples used in the current study. A very short steep time, in fact, was seen with the lower-protein barley (KWS Fantex) control. Einkorn, which had the smallest kernels, had the shortest mean steep time (49 h) of the wheat types. Einkorn cultivar TM23 which had a longer steep time than WB Alpine, was 1.3 percentage points higher in protein. The mean time for emmer was 52.6 h, but there was considerable variation between cultivars. Durum (62.5 h) and HRS wheat (66.4 h) had comparable mean steep times. The mean steep time for spelt was intermediate (56.3 h). Durum and HRS wheats are not commonly malted, and the longer steep times were likely associated with higher protein content and greater kernel vitreousness. Emmer wheat cultivars can also have higher vitreousness [38], which could partially explain the longer steep times. Einkorn and spelt lack vitreousness.

Maltsters select a target steep-out moisture to assure adequate hydration of the endosperm that is needed for uniform diffusion of enzymes and modification during germination. However, achieving a certain moisture level does not necessarily guarantee that endosperm has been fully hydrated, and for this reason hydration index (hydration quality) was also evaluated (Table 2). The steep index is based upon assessing the levels of hydration (e.g., 0, $\frac{1}{4}$, $\frac{1}{2}$, $\frac{3}{4}$ to full kernel) in 25 grains kernel after various time points. Individual kernels are scored from 0 to 4 points, based upon increasing hydration. The hydration index is the sum of all kernel scores. Considerable variation was seen between the wheat types. Both einkorn varieties were almost completely hydrated at 48 h, while the more vitreous HRS and durum wheats were generally not fully hydrated, even after 72 h. Spelt and emmer showed intermediate behavior. The emmer cultivar Vernal was removed from further study based upon a very long steep time (74.7 h) and slow hydration. These observations are interesting as the differing hydration patterns have implications for both commercial steeping and germination of these wheat types.

Figure 2 shows the appearance of various wheat types at 1 day of germination. Extensive growth of acrospires can be observed in both einkorn and emmer. Spelt and durum displayed good germination, but development of the acrospires was less advanced. Overall, the wheat samples displayed less rootlet development than for barley at 1 day. Some acrospires of einkorn samples also displayed a purple pigmentation.

### 3.3. Malt Extract, Soluble Protein, S/T Protein, FAN, Wort Color and Turbidity

High levels of fermentable extract are desired by brewers, as it is a determinant of the amount of malt needed in formulation. Wheat malts will often have higher levels than barley, as approximately 10–12% of the barley kernel is husk, which does not make a contribution to extract. In the current study, the highest levels of extract were seen in einkorn (84.8 %) and emmer (84.6%) followed by spelt (82.4%) and durum (82.1%) (Table 3). These values are quite similar to those previously reported for spelt [22], einkorn [21,37], and emmer [35,37]. Extract levels of HRS wheat varieties were all relatively low (<79%), which likely reflects high protein content and slower hydration and poor modification.

In the case of einkorn, emmer, spelt and durum, it is interesting to note that soluble protein levels followed the same rank as extract. In other words, while einkorn and emmer had higher levels of extract, a greater portion of this extract consisted of soluble protein, when compared to the other wheats. The mean soluble/total protein values (S/T) (Kolbach Index) for einkorn and emmer were both in fact above 52%, which suggests very extensive modification of the protein, and likely over-modification of the grain. The relationship between extract and soluble/protein was strong (r = 0.92) (Table 4). The S/T values in the current study are higher than those previously reported for einkorn and emmer malts [21,37], and likely reflect differences in the total protein of samples, and malting protocol used. In addition, the einkorn (TM23) and emmer (Lucile, ND Common) cultivars that exhibited the highest

soluble protein levels were also the same that had shown Fusarium/DON contamination in the malt. However, as non-infected cultivars also had high soluble protein levels, the impact of Fusarium infection, would seem to have been minimal.

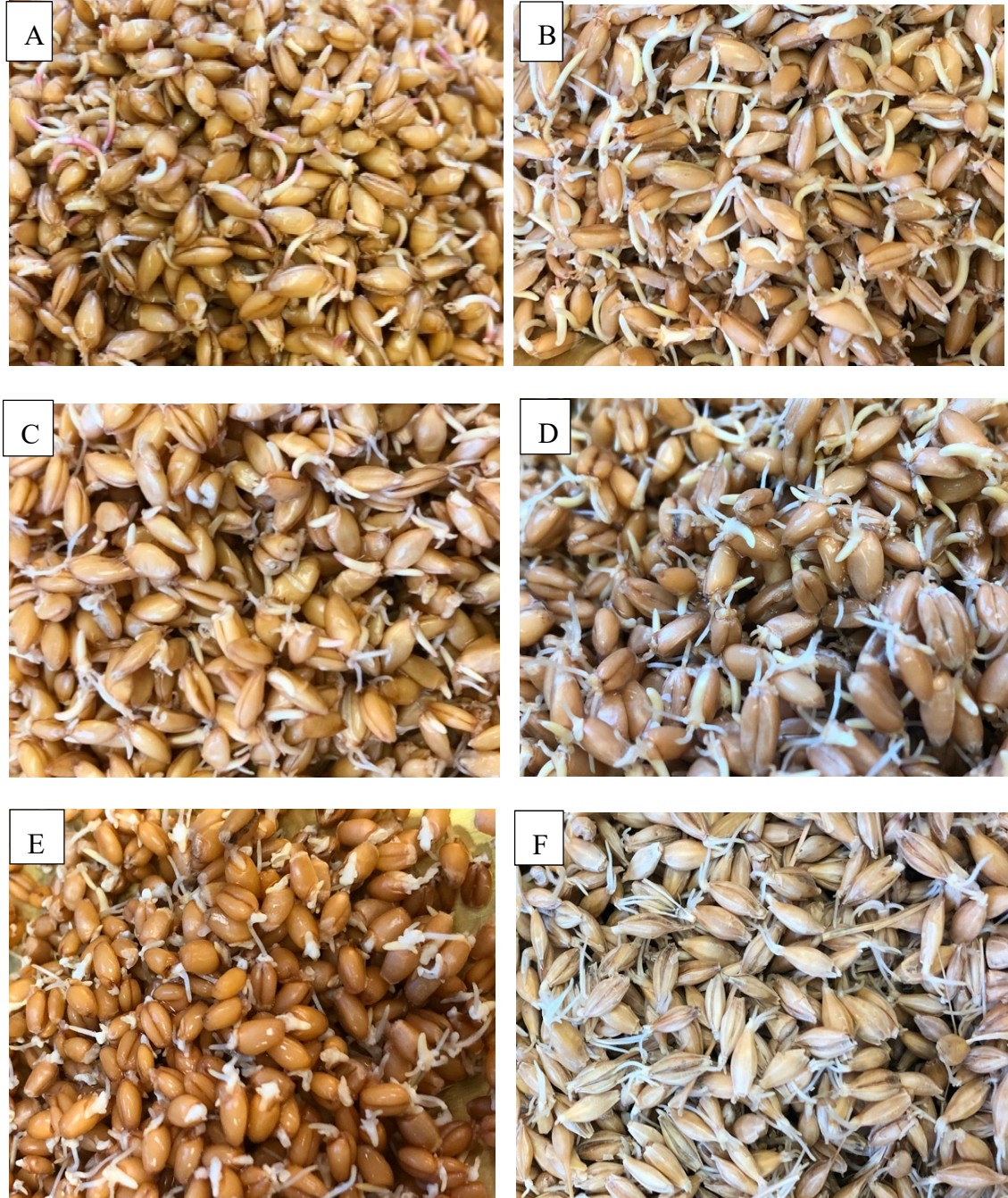

**Figure 2.** Appearance of micro-malts at 1 day of germination. (**A**) Einkorn (TM23), (**B**) emmer (ND Common), (**C**) spelt (CDC), (**D**) durum (Joppa), (**E**) HRS wheat (Barlow), and (**F**) barley (CDC Meredith).

**Table 3.** Malt quality parameters for wheat micro-malts and control malts.

| Sample | Variety | Extract (% Malt, db) | | Soluble Protein (%) | | S/T (%) | | FAN (mg/L) | | Color (SRM) | | Turbidity (NTU) | |
|---|---|---|---|---|---|---|---|---|---|---|---|---|---|
| | | Value | Average | Value | Average | Value | Average | Value | Average | Value | Average | Value | Average |
| | | | | | | Wheat Malt Samples | | | | | | | |
| Einkorn | TM23 | 84.0 | 84.8 ± 0.8 a | 7.2 | 7.1 ± 0.1 a | 51.4 | 52.9 ± 1.6 a | 166.1 | 180.2 ± 14 a | 4.2 | 3.9 ± 0.2 a | 28.9 | 23.0 ± 5.9 a |
| | WB Alpine | 85.7 | | 6.9 | | 54.5 | | 194.3 | | 3.7 | | 17.1 | |
| Emmer | Lucile | 84.0 | 84.6 ± 0.4 a | 6.3 | 6.2 ± 0.1 ab | 51.6 | 52.2 ± 0.5 a | 169.1 | 162.9 ± 5.1 a | 2.3 | 2.0 ± 0.4 b | 10.4 | 10.9 ± 1.7 bc |
| | ND Common | 84.5 | | 6.2 | | 53.2 | | 167.0 | | 2.4 | | 14.0 | |
| | Yaroslav | 85.3 | | 5.9 | | 51.9 | | 152.7 | | 1.2 | | 8.3 | |
| Spelt | CDC Zorba | 83.8 | 82.4 ± 0.7 b | 6.5 | 5.7 ± 0.3 b | 53.3 | 46.1 ± 3.7 ab | 167.0 | 151.2 ± 8.7 ab | 1.8 | 1.7 ± 0.07 b | 13.5 | 12.3 ± 0.6 bc |
| | 94-288 | 82.4 | | 5.4 | | 43.5 | | 136.9 | | 1.6 | | 12.0 | |
| | SK3P | 81.1 | | 5.4 | | 41.4 | | 149.9 | | 1.8 | | 11.3 | |
| Durum | ND Riverland | 81.3 | 82.1 ± 0.4 b | 4.5 | 4.3 ± 0.1 c | 41.3 | 40.7 ± 0.7 bc | 128.2 | 127.4 ± 3.4 b | 1.7 | 1.7 ± 0.05 b | 13.1 | 14.9 ± 1.6 b |
| | Joppa | 82.5 | | 4.1 | | 39.3 | | 121.0 | | 1.7 | | 13.6 | |
| | Divide | 82.6 | | 4.3 | | 41.6 | | 132.9 | | 1.6 | | 18.0 | |
| HRS | Barlow | 78.9 | 77.1 ± 0.6 c | 6.4 | 5.7 ± 0.3 b | 42.0 | 35.2 ± 2.5 c | 189.0 | 174.3 ± 10.3 a | 1.9 | 1.67 ± 0.1 b | 7.4 | 7.7 ± 0.6 c |
| | Glenn | 76.9 | | 5.8 | | 35.4 | | 179.0 | | 1.4 | | 9.4 | |
| | Linkert | 75.9 | | 4.9 | | 30.3 | | 144.0 | | 1.7 | | 8.0 | |
| | Sy Ingmar | 76.6 | | 5.7 | | 33.0 | | 185.0 | | 1.7 | | 6.3 | |
| | | | | | | Control Malt Samples | | | | | | | |
| Commercial wheat | | 86.0 | nc | 5.7 | nc | ne | nc | 181.8 | nc | 2.2 | nc | 12.9 | nc |
| CDC Meredith Barley Micro-Malt | | 78.9 | | 5.7 | | 39.9 | | 267.2 | | 2.6 | | 9.6 | |
| KWS Fantex Barley Micro-Malt | | 81.6 | | 4.9 | | 52.6 | | 237.2 | | 2.9 | | 20.9 | |
| Commercial Barley (A) | | 82.5 | | 4.7 | | ne | | 211.6 | | 1.9 | | 9.0 | |
| Commercial Barley (B) | | 82.4 | | 5.0 | | ne | | 226.8 | | 2.0 | | 5.6 | |

Notes: S/T—soluble protein; FAN—free amino nitrogen; SRM—Standard Reference Method; NTU—Nephelometric Turbidity Units; nc—not calculated, commercial wheat and barley malts were not considered in the statistical analysis; ne—not evaluated; different letters on the average values within columns signify significant differences ($p < 0.05$).

**Table 4.** Pearson's correlation coefficients for wheat micro-malts ($p < 0.05$).

| | Malt Extract | Protein | DP | α-Amylase | Sol Protein | S/T | FAN | Viscosity | Color | β-Glucan | Protease | Xylanase | α-Amylase | β-Amylase | AX HMW | Ferulic Acid | Total PA |
|---|---|---|---|---|---|---|---|---|---|---|---|---|---|---|---|---|---|
| Malt Extract | 1 | −0.781 | −0.649 | 0.251 | 0.330 | 0.920 | −0.075 | 0.221 | 0.438 | 0.130 | 0.545 | −0.343 | −0.629 | −0.903 | −0.163 | 0.737 | 0.749 |
| Protein | | 1 | 0.613 | 0.261 | 0.299 | −0.541 | 0.594 | 0.064 | 0.015 | −0.228 | −0.511 | −0.024 | 0.609 | 0.769 | 0.359 | −0.440 | −0.483 |
| DP | | | 1 | 0.160 | −0.061 | −0.499 | 0.317 | −0.323 | −0.523 | −0.045 | −0.693 | 0.155 | 0.779 | 0.893 | 0.078 | −0.615 | −0.631 |
| α-Amylase | | | | 1 | 0.817 | 0.542 | 0.791 | 0.183 | 0.347 | 0.091 | 0.008 | −0.486 | 0.346 | 0.008 | 0.342 | 0.121 | 0.093 |
| Sol Protein | | | | | 1 | 0.635 | 0.822 | 0.355 | 0.647 | 0.045 | 0.015 | −0.523 | 0.043 | −0.205 | 0.276 | 0.388 | 0.357 |
| S/T | | | | | | 1 | 0.253 | 0.236 | 0.523 | 0.223 | 0.414 | −0.468 | −0.412 | −0.774 | −0.047 | 0.680 | 0.686 |
| FAN | | | | | | | 1 | 0.171 | 0.408 | −0.122 | −0.169 | −0.281 | 0.451 | 0.243 | 0.345 | −0.022 | −0.065 |
| Viscosity | | | | | | | | 1 | 0.472 | 0.041 | 0.080 | −0.625 | −0.388 | −0.295 | 0.723 | 0.348 | 0.344 |
| Color | | | | | | | | | 1 | −0.258 | 0.482 | −0.164 | −0.538 | −0.545 | 0.377 | 0.785 | 0.768 |
| β-glucan | | | | | | | | | | 1 | −0.125 | −0.346 | 0.025 | −0.104 | 0.043 | −0.164 | −0.103 |
| Protease | | | | | | | | | | | 1 | 0.036 | −0.527 | −0.627 | 0.008 | 0.620 | 0.626 |
| Xylanase | | | | | | | | | | | | 1 | 0.046 | 0.256 | −0.296 | −0.175 | −0.165 |
| α-amylase | | | | | | | | | | | | | 1 | 0.822 | −0.027 | −0.802 | −0.820 |
| β-amylase | | | | | | | | | | | | | | 1 | 0.147 | −0.775 | −0.793 |
| AX HMW | | | | | | | | | | | | | | | 1 | 0.095 | 0.101 |
| Ferulic acid | | | | | | | | | | | | | | | | 1 | 0.996 |
| Total PA | | | | | | | | | | | | | | | | | 1 |

It is interesting to note that, despite higher soluble protein in emmer and einkorn, the values for FAN were still below those observed for the control barley malts (Table 3). Malts from all wheat types and the wheat control malt were all lower in FAN than the barley malts. However, within the wheat samples a strong relationship was observed between FAN and soluble protein (r = 0.82) (Table 4). With the exception of einkorn, there were no significant differences in wort color between the wheat types, and color was similar to slightly lower than observed for the control wheat and barley malts. However, the average einkorn wort color (3.9 SRM) was significantly darker that seen for other wheat types (1.7–2.0 SRM). Einkorn worts were also significantly more turbid. This color difference was not noted in previous studies on einkorn and emmer malts [34,37]. While einkorn is known to have higher levels of carotenoid pigments, these are not water soluble.

*3.4. Malt Enzymes*

Standard malt analyses generally only report $\alpha$-amylase and diastatic power (DP) determined according to methods of the American Society of Brewing Chemists (ASBC) or European Brewery Convention (EBC). Alpha-amylase is an endoenzyme that hydrolases $\alpha$-1,4 glycosidic linkages in starch, which results in a reduction in molecular weight and viscosity. Diastatic power (DP) is a measure of all the enzymes which hydrolyze starch, but is thought to largely reflect $\beta$-amylase [46], because of its high turnover number (kcat). Beta-amylase is an exoenzyme, which acts on the non-reducing end of starch and dextrins to produce maltose.

Results in Table 5 show that there was very little difference between wheat types for $\alpha$-amylase when determined by the ASBC method. Mean levels, when averaged across cultivars for einkorn, emmer and spelt (43–49 DU), were comparable to those observed in the commercial wheat malt (45 DU), but lower that those observed for the barley malt controls. In contrast, levels of DP in emmer (168 °ASBC) and spelt (167) were comparable to slightly higher than most of the barley control malts. The DP of barley control KWS Fantex (44) was quite low, but this sample was also very low in total grain protein. Einkorn was the exception in the current study, and the mean DP (116 °ASBC) was significantly lower than all other wheat types (167–207). Mayer et al. [35] did note lower DP in a single einkorn sample when compared to emmer. In barley, total grain protein is positively related to DP levels [55], and DP divided by protein (DP/N) reveals anomalies in this relationship. Values for most wheat types were similar (DP/N: 12.7–14.2), but the value for einkorn (8.7) was significantly lower (Table 5). Einkorn also had a significantly lower mean $\beta$-amylase activity (12.7 units/g) when compared to the other wheat types (22.6–49.4 units/g). Levels were highest in HRS wheat, which also had the highest grain protein levels. Beta-amylase levels of the other ancient wheats were similar to the commercial wheat malt, but higher than seen in barley. Beta-amylase is found in the endosperm, and higher levels in wheat, to a degree, reflect lack of a husk, and proportionally more endosperm/g.

Megazyme test kits were also used to determine the development of $\alpha$- and $\beta$-amylase, protease and endoxylanase activities over the course of the malting process (Table 5 and Figure 3D). Not surprisingly, levels of $\alpha$-amylase in the wheat grain and control barley samples were zero to negligible. In all cases, activity began to increase between 1 and 2 days of germination and reached maximum levels at 4 days. Levels at 4 days were highest in barley (261 units/g) and HRS wheat (190), and lowest in einkorn (95) and durum (107). Almost 40% of the activity present in barley at 4 days was lost following kilning. Kilning losses in the wheat samples were not as great (e.g., 21% in einkorn). In terms of $\beta$-amylase, virtually no change was seen over the course of germination for any of the samples. This is as cysteine was included in the extraction buffer, resulting in the conversion of insoluble (inactive) $\beta$-amylase to soluble (active). In retrospect, it may have been more informative to conduct the extraction in the absence of cysteine, so as to get a measure of increase in the soluble form over germination. Following kilning, a 47% decline in $\beta$-amylase was observed in the barley samples. This was not unexpected as barley $\beta$-amylases are known to be heat liable. The decline in wheat samples was less pronounced, and activity appeared to increase slightly in einkorn.

**Table 5.** Enzyme activities in wheat micro-malts and controls.

| Sample | α-Amylase (DU) (Official ASBC Method) | α-Amylase (U/g db) (Test Kit) | DP (ASBC) (Official ASBC Method) | DP/N | β-Amylase (U/g db) (Test Kit) | Protease (U/g db) (Test Kit) | Xylanase (mU/g db) (Test Kit) |
|---|---|---|---|---|---|---|---|
| | | | Wheat Malt Samples | | | | |
| Einkorn | 49.5 ± 3.1 a | 75.3 ± 15.3 c | 116.0 ± 4.0 c | 8.7 ± 0.7 b | 12.7 ± 0.0 c | 1.8 ± 0.05 a | 2.3 ± 1.1 bc |
| Emmer | 46.6 ± 2.7 a | 126.3 ± 9.2 b | 168.1 ± 7.2 ab | 14.2 ± 0.5 a | 22.6 ± 0.5 bc | 1.6 ± 0.1 a | 2.7 ± 1.3 bc |
| Spelt | 43.3 ± 5.8 ab | 112.7 ± 9.5 b | 167.3 ± 10.9 ab | 13.4 ± 1.1 a | 27.2 ± 1.4 b | 1.3 ± 0.03 b | 1.9 ± 0.2 c |
| Durum | 35.0 ± 0.9 b | 99.8 ± 6.8 bc | 145.2 ± 12.6 bc | 13.7 ± 1.3 a | 25.0 ± 2.1 b | 1.8 ± 0.2 a | 6.4 ± 0.4 a |
| HRS | 44.6 ± 2.6 ab | 164.7 ± 10.1 a | 206.8 ± 18.7 a | 12.7 ± 0.8 a | 49.4 ± 4.4 a | 1.3 ± 0.0 b | 4.4 ± 0.8 ab |
| | | | Control Malt Samples | | | | |
| Commercial Wheat | 44.5 | 107.0 | 192.2 | ne | 27.2 | 1.3 | 1.1 |
| CDC Meredith Barley Micro-Malt | 65.4 | 177.7 | 137.6 | 9.63 | 16.1 | 1.6 | 17.1 |
| KWS Fantex Barley Micro-Malt | 66.4 | 139.1 | 44.2 | 4.70 | 3.0 | 2.3 | 7.2 |
| Commercial Barley (A) | 59.9 | 154.9 | 132.8 | ne | 12.0 | 1.5 | 5.5 |
| Commercial Barley (B) | 55.0 | 124.2 | 148.1 | ne | 13.7 | 1.7 | 3.5 |

Notes: DP—diastatic power; DP/N—diastatic power/grain protein content; ne—not evaluated; different letters on the average ± std values within columns signify significant differences ($p < 0.05$). Commercial wheat and barley malts were not considered in the statistical analysis.

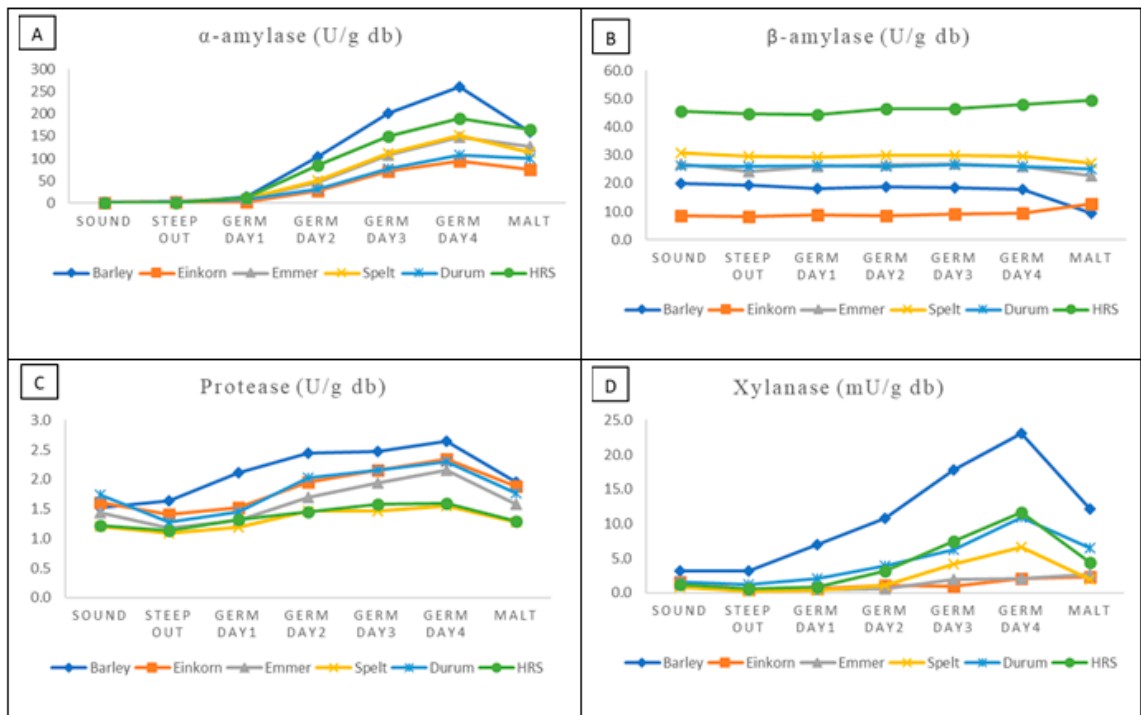

**Figure 3.** Change in enzyme activities over the course of the malting process: α-amylase (**A**), β-amylase (**B**), protease (**C**) and xylanase (**D**).

Endo-protease activity was measured using an azurine-crosslinked casein substrate, and it should be noted that this substrate may not be completely reflective of the activities of all proteases in germinating cereals on their native storage proteins. As anticipated some endoprotease activity was observed in the sound grain of all samples (1.2–1.7 units/g) (Table 5). In general, activity began increasing following steep out (Figure 3). Barley protease activity increased by 1.7 fold following 4 days of germination, while einkorn and emmer activities increased by 1.5 fold. Activity loses following kilning for barley, einkorn and emmer were all approximately 20–25%. Protease activities in the kilned malts were highest in barley and einkorn, and lowest in HRS wheat and spelt. However, the overall variation between wheat types for protease activity was somewhat limited (1.3–1.9 units/g).

Endo-xylanase activity was measured in the malt samples as levels play a role in the solubilization and degradation of arabinoxylans, which in term may influence wort viscosity. The highest levels of xylanase in kilned wheat malts were observed in HRS (4.4 units/g) and durum (6.4), while the lowest were observed in the ancient wheats (1.9–2.7) (Table 5). Overall, the levels in the micro-malts (wheat and barley) were considerably higher than those of the commercial controls, which likely reflects differences in the two processes. In barley, activity was observed to increase following steep out, and at 4 days had reached 23.1 units/g, which represented a 7.2-fold increase (Figure 3). The rate of increase in the wheat samples was slower, as were the maximum levels at 4 days. Four-day levels in durum (10.9 units/g) and HRS (11.6) were highest followed by spelt (6.6). Very low levels were observed in einkorn (2.4) and emmer (2.2), even after four days. Approximately 47% of the activity developed during the germination of barley was lost in kilning. Similar losses were seen in durum and spring wheat.

### 3.5. Wort β-Glucan, Arabinoxylan and Viscosity

Beta-glucans are the major endosperm cell wall polysaccharide in barley and oats [36]. However, in wheat and rye, arabinoxylans (AX) serve this function. AX and β-glucan levels are directly related to the viscosity [56], and consequently lautering and filterability in the brewery. In addition, it has been

suggested that AX can have a positive effect on quality of beer, such as improving sensorial properties and stabilizing the foam [57]. Wort viscosities higher than 1.5 cP have historically been used as an indicator of lautering and filtration problems [37,58], but today, the measurement of wort β-glucan is more common. AX levels, however, are not routinely measured.

Not surprisingly, the levels of β-glucan in all wheat samples were quite low (Table 6). Barley levels were relatively high, but our micro-malting procedure tends to result in under-modification of β-glucan. The commercial barley malts were within normal range. On the other hand, levels of total AX and high-MW AX in the wheat sample worts were much higher than β-glucan. Total AX is based on measurement of all xylose and arabinose in the wort following acid hydrolysis. As such, the source could vary from monomeric arabinose and xylose to AX polysaccharides. In contrast, high-MW AX is based on hydrolysis of materials precipitated from the wort with 80% ethanol. Examination of the data shown in Table 6 shows that values for high-MW AX were almost always lower than those for AX. On average, levels of high-MW AX were greatest in einkorn and spelt. Levels also trended higher in HRS wheat, but as this sample was not well modified, it might reflect more limited degradation of water-soluble AX. The relationships seen between levels of xylanase and high-MW AX were poor (r = −0.30) (Table 4). In contrast, the relationship between wort viscosity and high-MW AX was strong (0.72). Wort viscosity might be an issue for einkorn and spelt (Table 6). Einkorn had previously been shown to have low levels of endoxylanase activity (Table 5). Our results for einkorn viscosity were lower than those reported by Marconi et al. [38], but results for spelt were similar to those of Munoz-Insa et al. [36]

### 3.6. Wort Phenolic Acids

Phenolic acids in wort and beer are associated with flavor, non-biological stability, color and other characteristics [59]. As an example, some yeasts are able to convert ferulic acid to 4-vinyl guaiacol, which has strong clove aroma and flavor [60]. Phenolic acids in cereal grains are often bound with cell wall polysaccharides, with some being liberated during germination [61]. In this study, we evaluated the levels of soluble free and bound phenolic acids present in the laboratory worts (Table 7). Those reported as bound are still associated with soluble saccharides. Results are divided between hydroxycinnamic and hydroxybenzoic acids. Mean values ± ste (standard error) were calculated for each grain type. Total phenolic acids were calculated as the sum of all individual acids (free and bound) measured. The distribution of individual phenolic acids is also shown in Figure 4.

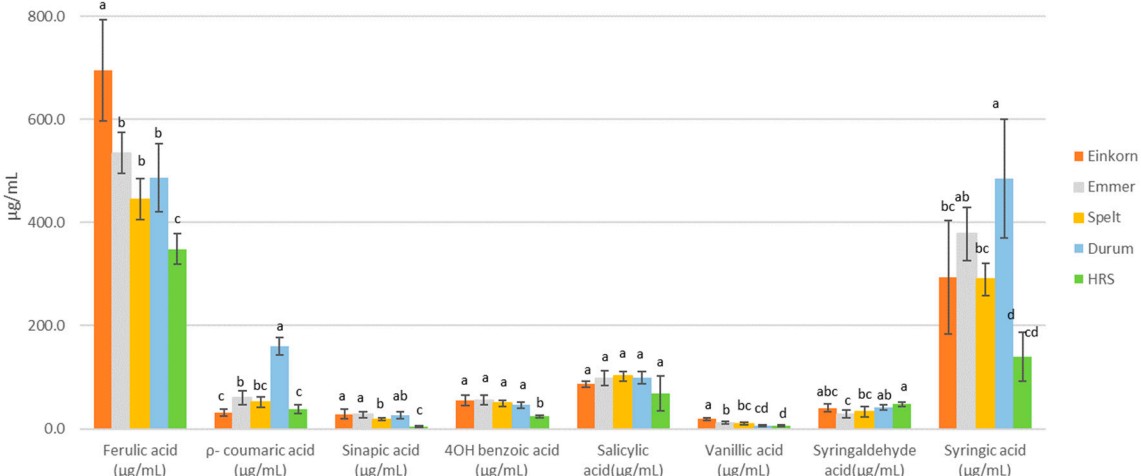

**Figure 4.** Phenolic acids measured in wort from wheat micro-malts. Values for individual acids represent the sum of free and bound forms. Different letters for wheat type mean within a compound signify significant differences (*p* < 0.05).

**Table 6.** Wort β-glucan, arabinoxylan and viscosity for wheat micro-malts and control malts.

| Sample | Variety | β-glucan (mg/L) | | Total AX (g/L) | | A/X (in Total AX) | | HMW AX (g/L) | | A/X (in HMWAX) | | Viscosity (mPa.s] | |
|---|---|---|---|---|---|---|---|---|---|---|---|---|---|
| | | Value | Average | Value | Average | Value | Average | Value | Average | Value | Average | Value | Average |
| *Wheat Malt Samples* | | | | | | | | | | | | | |
| Einkorn | TM23 | 26.3 | 25.4 ± 0.9 b | 2.18 | 2.6 ± 0.3 a | 1.03 | 1.0 ± 0.03 c | 1.76 | 2.1 ± 0.3 a | 1.06 | 1.0 ± 0.04 c | 1.8 | 1.95 ± 0.1 a |
| | WB Alpine | 24.5 | | 2.95 | | 0.97 | | 2.53 | | 0.97 | | 2.1 | |
| Emmer | Lucile | 28.4 | 28.6 ± 0.1 b | 1.13 | 1.3 ± 0.06 c | 1.45 | 1.4 ± 0.0 a | 1.08 | 1.2 ± 0.1 c | 1.40 | 1.4 ± 0.24 a | 1.5 | 1.6 ± 0.03 c |
| | ND Common | 28.8 | | 1.3 | | 1.43 | | 1.47 | | 1.41 | | 1.6 | |
| | Yaroslav | 28.6 | | 1.34 | | 1.45 | | 1.04 | | 1.48 | | 1.6 | |
| Spelt | CDC Zorba | 43.4 | 37.5 ± 3.5 a | 1.98 | 2.2 ± 0.1 ab | 0.96 | 1.0 ± 0.0 c | 1.97 | 1.8 ± 0.2 ab | 1.04 | 1.0 ± 0.02 c | 1.7 | 1.8 ± 0.1 ab |
| | 94-288 | 31.2 | | 2.32 | | 0.95 | | 2.07 | | 0.99 | | 2.0 | |
| | SK3P | 37.9 | | 2.29 | | 0.98 | | 1.38 | | 0.97 | | 1.8 | |
| Durum | ND Riverland | 36.5 | 28.6 ± 4.0 b | 1.82 | 1.7 ± 0.3 bc | 1.11 | 1.2 ± 0.05 b | 1.79 | 1.4 ± 0.2 bc | 1.16 | 1.2 ± 0.04 b | 1.6 | 1.6 ± 0.03 c |
| | Joppa | 23.5 | | 1.14 | | 1.27 | | 1.12 | | 1.28 | | 1.5 | |
| | Divide | 25.8 | | 2.01 | | 1.20 | | 1.42 | | 1.17 | | 1.6 | |
| HRS | Barlow | 27.0 | 26.8 ± 1.6 b | 1.85 | 2.2 ± 0.2 ab | 1.00 | 1.0 ± 0.02 c | 1.58 | 1.8 ± 0.1 ab | 1.00 | 1.0 ± 0.0 c | 1.7 | 1.7 ± 0.03 bc |
| | Glenn | 31.0 | | 2.03 | | 0.95 | | 1.63 | | 1.01 | | 1.6 | |
| | Linkert | 25.0 | | 2.54 | | 1.03 | | 2.00 | | 1.00 | | 1.7 | |
| | Sy Ingmar | 24.0 | | 2.24 | | 0.99 | | 1.94 | | 0.99 | | 1.6 | |
| *Control Malt Samples* | | | | | | | | | | | | | |
| Commercial Wheat | | 43.8 | nc | 1.82 | nc | 0.90 | nc | 1.61 | nc | 0.96 | nc | 1.6 | nc |
| CDC Meredith Barley Micro-Malt | | 295.7 | | 1.33 | | 0.89 | | 0.66 | | 1.01 | | 1.4 | |
| KWS Fantex Barley Micro-Malt | | 314.3 | | 1.08 | | 1.03 | | 0.67 | | 1.10 | | 1.6 | |
| Commercial Barley (A) | | 104.9 | | 1.17 | | 1.00 | | 1.01 | | 0.91 | | 1.5 | |
| Commercial Barley (B) | | 59.6 | | 1.12 | | 1.05 | | 0.85 | | 1.01 | | 1.5 | |

Note: AX—arabinoxylan; A/X—arabinose: xylose ratio; HMW AX—high molecular weight of arabinoxylan; nc—not calculated barley was not considered in the statistics; different letters on the columns means significant difference ($p < 0.05$).

**Table 7.** Phenolic acids in wort.

| Classes | Compounds (µg/mL) | Forms- | Wheat Malt Samples | | | | | | Control Malt Samples | | | |
|---|---|---|---|---|---|---|---|---|---|---|---|---|
| | | | Einkorn | Emmer | Spelt | Durum | HRS | Wheat | CDC Meredith | KWS Fantex | Barley (A) | Barley (B) |
| Hydroxycinnamic acid | Ferulic acid | Free | 205.8 ± 2.5 a | 120.8 ± 17.0 bc | 73.4 ± 7.0 d | 153.0 ± 13.9 b | 106.7 ± 7.6 cd | 58.9 | 266.5 | 261.1 | 143.7 | 109.7 |
| | | Bound | 489.5 ± 71.8 a | 413.8 ± 16.7 ab | 371.8 ± 23.0 b | 333.6 ± 24.2 b | 241.9 ± 9.8 c | 344.2 | 456.0 | 364.8 | 412.1 | 308.0 |
| | ρ-coumaric acid | Free | 3.5 ± 2.4 b | 25.4 ± 3.9 b | 17.9 ± 2.9 b | 108.6 ± 12.2 a | 16.8 ± 3.2 b | 12.6 | 64.8 | 54.8 | 76.2 | 48.1 |
| | | Bound | 28.1 ± 2.5 bc | 34.4 ± 4.1 b | 34.4 ± 4.2 b | 51.2 ± 2.4 a | 21.2 ± 2.8 c | 24.1 | 31.4 | 37.9 | 41.4 | 26.6 |
| | Sinapic acid | Free | 4.2 ± 0.9 ab | 2.4 ± 0.1 bc | 2.6 ± 0.6 b | 5.6 ± 1.1 a | 0.6 ± 0.3 c | 0.1 | 0.8 | 3.4 | 9.9 | 8.4 |
| | | Bound | 24.6 ± 5.5 a | 25.0 ± 3.4 a | 15.5 ± 0.9 b | 20.8 ± 2.6 ab | 4.4 ± 0.9 c | 18.0 | 23.0 | 89.1 | 131.0 | 62.8 |
| Hydroxybenzoic acid | 4OH benzoic acid | Free | 34.3 ± 5.5 a | 35.4 ± 4.9 a | 30.7 ± 2.8 a | 28.6 ± 2.6 a | 15.0 ± 0.8 b | 32.7 | 4.7 | 5.9 | 3.8 | 3.3 |
| | | Bound | 20.8 ± 1.3 a | 20.6 ± 0.9 a | 18.8 ± 0.9 a | 17.7 ± 0.6 a | 9.0 ± 0.9 b | 15.5 | 15.7 | 9.4 | 8.7 | 13.3 |
| | Salicylic acid | Free | 23.3 ± 0.7 a | 29.0 ± 2.6 a | 41.9 ± 10.6 a | 28.7 ± 2.8 a | 38.7 ± 6.75 a | 33.2 | 44.3 | 99.8 | 54.4 | 48.6 |
| | | Bound | 63.4 ± 3.5 a | 69.1 ± 5.8 a | 60.5 ± 5.5 a | 71.1 ± 4.1 a | 29.6 ± 11.5 b | 65.8 | 14.4 | 25.1 | 22.9 | 58.7 |
| | Vanillic acid | Free | 13.3 ± 0.9 a | 7.1 ± 0.9 b | 6.5 ± 0.7 bc | 4.7 ± 0.4 cd | 3.7 ± 0.3 d | 4.4 | 22.6 | 23.5 | 24.4 | 16.9 |
| | | Bound | 5.8 ± 1.1 a | 4.7 ± 0.9 a | 3.7 ± 1.0 ab | 1.8 ± 0.8 b | 2.3 ± 0.25 b | 2.8 | 37.6 | 33.2 | 41.5 | 31.1 |
| | Syringaldehyde acid | Free | 10.1 ± 2.1 b | 7.2 ± 0.8 b | 3.3 ± 0.1 c | 8.1 ± 0.2 b | 14.7 ± 0.9 a | 4.7 | 1.9 | 3.1 | 3.0 | 2.9 |
| | | Bound | 30.3 ± 3.0 ab | 21.2 ± 3.8 b | 29.6 ± 5.8 ab | 33.3 ± 2.9 a | 33.0 ± 1.5 a | 26.9 | 1.2 | 2.3 | 0.9 | 0.8 |
| | Syringic acid | Free | 8.7 ± 5.4 c | 28.6 ± 3.2 ab | 21.3 ± 3.4 b | 34.3 ± 3.1 a | 10.2 ± 0.9 c | 5.2 | 1.8 | 2.2 | 2.5 | 1.0 |
| | | Bound | 284.9 ± 72.1 b | 349.2 ± 26.9 ab | 269.0 ± 15.9 b | 450.6 ± 62.9 a | 129.2 ± 23.3 c | 164.6 | 6.9 | 5.7 | 9.8 | 6.2 |
| Sum of analyzed Phenolic acids | | | 1251 ± 1.4 ab | 1194 ± 41.6 b | 1000 ± 34.6 c | 1352 ± 52.5 a | 677 ± 33 d | 814.0 | 994.0 | 1021.0 | 986.0 | 746.0 |

Note: different letters on the average ± std values within columns signify significant difference ($p < 0.05$). Commercial wheat and barley malts were not considered in the statistical analysis.

In terms of total phenolic acids, einkorn, emmer and durum wheats trended towards higher levels (1194–1352 µg/mL, Table 7). The value for HRS wheat (677 µg/mL) was the lowest, and likely partially reflected the poor modification of these samples. Values for the two barley controls were comparable to those of spelt.

Ferulic acid was the most abundant hydroxycinnamic acid in all the samples (Table 7, Figure 4). The commercial white wheat malt followed the same pattern. The barley controls were both higher than the wheat types, although barley was not included in the statistical analysis. With the exception of spelt, approximately 30% of the total ferulic acid in the wheat samples was in the free form. Values were a little higher for the barley controls, with 37–42% in the free form. Among the wheat types, einkorn had the highest level of free ferulic acid followed by durum, emmer, HRS, and spelt. Levels of p-coumaric and sinapic acids were, in almost all cases, much lower than those seen for ferulic acid. The only exception was for durum, where the level of p-coumaric acid was approximately 70% of the free ferulic acid. For the hydroxybenzoic acids, syringic acid was the most abundant (Table 7), and represented 20 to 36% of the total phenolic acids measured in the wheat samples. Barley has had very low levels of syringic acid (9% of total phenolic acids).

The levels of both ferulic and total phenolic acids (TPA) showed a strong relationship with wort color (r = 0.76) (Table 4). While some of this may be attributable to direct effects, there was also a strong relationship between both ferulic and total phenolic acids and S/T protein (r = 0.68), which suggests that increasing modification was also a factor.

## 4. Discussion

Samples utilized for the current study demonstrated that a potential problem in the use of ancient wheats for malting is mechanical damage caused by the dehulling process. This was particularly problematic with einkorn, which had been previously reported [10]. Visibly broken kernels in the current study were removed by hand, but this is not practical. Several potential dehulling options for ancient wheats were discussed in a report by Baker [51]. This report also mentioned brewers and distillers as a potential market for hulled wheats, but that maltsters require dehulling as the hulls may impart off flavors. Mayer et al. [35] and Marconi et al. [38] reported the malting of hulled einkorn and emmer, and that acceptable beer could be prepared. However, the use of hulled wheats in malting could potentially present microbiological and food safety concerns and should be investigated. This is as it has been demonstrated that Fusarium and DON content can be higher in the glumes than in the grain [62], and that mycotoxins can increase during malting [63].

Additional quality issues with the hulled wheats were germination and protein. In the current study the germinative energy of most hulled wheat samples was below the 95% that is normally required for barley. Samples with very poor germination were not malted. Reduced germination in the current study could have been caused by damage in harvest or dehulling. Unlike barley growers, wheat growers are not normally concerned with delivering grain in a living state. Growers producing hulled wheats for the malting market will thus need to be cognizant of potential damage to germination that can occur during harvest, dehulling, and in improper grain storage. The protein content of wheats used in the current study (11.4 to 17.3%) was also higher than that normally used for wheat malt. Briggs stated that for brewers malt, wheats with <1.9% total nitrogen (≈10.8% protein) should be selected [37]. Considerable range in protein content has been reported for both emmer [64] and spelt cultivars [65], but einkorn trends towards high levels [66]. Those wishing to utilize these grains in malting may be able to control protein content to a degree through cultivar selection and agronomic practices. However, cultivar choice is generally more limited than for barley or common wheat.

The higher protein content, as well as vitreousness in some cases, undoubtedly contributed to the longer steep times observed for the wheat samples in the current study. Another factor is the steep regime employed in our micro-malting protocol. As our system lacks temperature control during air-rests we utilize relatively long immersions (11 h). It is well known that shorter immersions and longer air rests will speed hydration [67]. According to Briggs, wheat malts were traditionally steeped

to only 42% moisture and germinated at temperatures <16 °C to reduce microbial issues, but better modification can be achieved at 45% moisture [37]. In the current study, einkorn samples were well hydrated at 48 h, even though they had not yet reached 45% moisture. When steeped to 45% and germinated for 4 days, the samples were over-modified, as evidenced by high soluble and S/T protein, and malt losses above 10% (Table 2). As such, einkorn malt quality might benefit from steep-out moistures <45%, and possibly from shorter germination times. However, Sachambula et al. [21] previously reported that 45% moisture and 4 days of germination were the best conditions for einkorn malt quality. Muñoz-Insa et al. [36] reported that optimal quality of spelt malt was achieved with 47% of moisture, and 5 days of germination. Harder or more vitreous wheats (e.g., durum, HRS) may benefit from shorter immersions, and if still not fully hydrated after steeping the maltster may wish to consider longer germination times, and addition of more water during germination. The results of the current study, as well as those of previous studies [21], clearly indicate that ideal malting conditions will change for the different wheat types. Further, the variations (e.g., protein, hardness, and kernel size) due to genotype and growing environment have not really been addressed, as most studies to date have only used single to a very limited number of cultivars.

Hierarchical cluster analysis (HCA) was utilized to determine the similarity of the wheat micro-malts in terms of basic grain and malt quality, and results shown in Figure 5 demonstrated that the different wheat types exhibited distinct characteristics. Emmer and einkorn appeared to be most similar, followed by spelt and durum. The overall characteristics of the HRS sample were most distinct. This is interesting as spelt and HRS are both hexaploid wheats. However, in the current study, there was considerable difference in protein content between the two types, and the HRS samples were also likely much more vitreous.

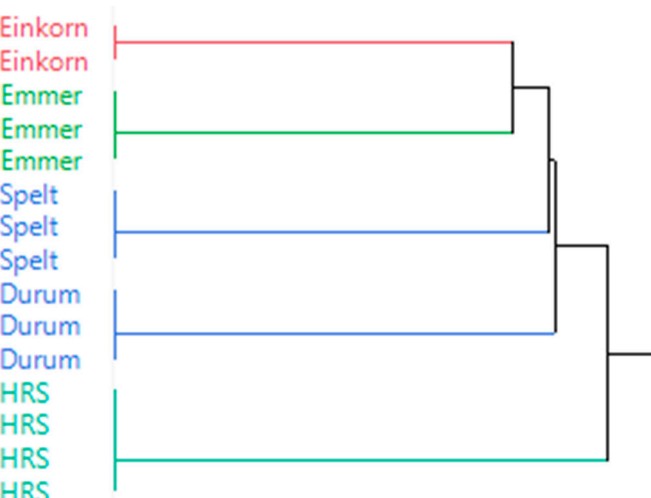

**Figure 5.** Hierarchical cluster analysis (HCA) applied to analyze the similarity of the wheat micro-malts.

Results of the current and previous studies have generally indicated that einkorn, emmer and spelt malts have potential for use in brewing. This is particularly true, if one considers that these grains are most likely to be used as a component of a grist bill, rather than the sole grain. Extract and $\alpha$-amylase levels of all ancient wheat malts were within an acceptable range. Levels of DP in spelt and emmer were actually higher than those observed in the barley control malts. While DP was observed to be significantly lower in einkorn, this should not present a problem if it is used in conjunction with other enzymatic malts. One negative issue was the higher soluble and S/T protein that was observed with einkorn and emmer. This may be a result of higher grain protein and malting conditions of the current study. Einkorn also appeared to yield wort that was darker and more turbid, which could be a consideration for its use in some beer styles. However, testing of more cultivars across multiple growing locations would be needed to determine if higher soluble protein and color are true characteristics of these wheat types.

Given the fact that the production of einkorn, emmer and spelt is limited, and is often from organic production, cost will likely be considerably higher than for common wheat. As such, their use in brewing will likely be limited to specialty products. While the novelty of ancient wheats and the associated narrative can provide some market justification, promotion can also be based upon actual or presumed health benefits. Phenolic acids [68], polyphenols and important minerals [12] have all been promoted as possible health benefits of using ancient wheats in brewing. One must also consider whether malting provides any enhancement in these characters, as these grains can also be utilized as adjuncts. From the health benefit point of view, β-glucan and AX are considered dietary fiber, which is beneficial to lowering cholesterol, attenuating type II diabetes, and enhancing minerals absorption [67]. In the current study both einkorn and spelt exhibited relatively higher levels of AX and high-MW AX. Previous research has been demonstrated that levels of wort AX increase with increasing germination time [22], and although not shown as part of current study, our work with these samples also demonstrated that soluble AX increases from sound grain to malt.

Einkorn and emmer had high levels of free and soluble-bound (conjugated) ferulic acid when compared to the other wheat types (Table 7). Although the data were not presented, as part of the current study, we observed that both free and conjugated ferulic acid increased with the malting of all wheat types, except spelt. In wort, soluble conjugated forms were present in greater amounts than the free forms, with the exception of ρ-coumaric acid in barley and durum worts. Studies have shown that hydroxycinnamic acids have higher antioxidant efficiency than hydroxybenzoic acids because of the carboxylic group [48]. Resultant worts could have strong antioxidant capacities, and can act as a free-radical scavengers and transition metal chelators [69], which has implications for several chronic diseases [13,14]. In vivo experiments have also shown that soluble free and conjugated ferulic acid are easily absorbed in the small intestine [70,71].

## 5. Conclusions

Recent years have seen dramatically increased interest in the ancient wheats, and in terms of brewing, beers utilizing either einkorn, emmer or spelt are already available in the marketplace. The results of the current study and several previous studies have demonstrated malts of acceptable quality can be prepared from these grains. However, as these grains are not widely available, a limitation of past work has been the small numbers of cultivars tested, often from different growing environments. The current study utilized several cultivars of each wheat type from the same growing environment. Results showed that the wheat types exhibited distinct behaviors in terms of grain and malt quality parameters, with einkorn and emmer being more similar than spelt. Although not extensively investigated as part of the current study, malting conditions will likely be needed to be optimized for each grain type, as well for the variations (e.g., kernels size, protein) seen between growing environments. As an example, einkorn, with its smaller kernels, is faster to hydrate and modify than either spelt or emmer.

Dehulling of these wheats, especially einkorn, can result in kernel damage and reduced germination. The malting of grain without dehulling has been proposed as an option. However, this could pose potential food safety issues, as fungi and mycotoxins can be present in the glumes, as well as on the grain itself. As DON is relatively common and was detected in several of the samples in the current study, we feel that investigation of the microbiological aspects of malting these wheats with the glumes should be further investigated.

The use of these grains in brewing will likely be limited to specialty products or limited releases, as supply is very limited when compared to common wheat. Consumer interest in ancient grains and their potential health benefits is likely a driving factor, as is the desire of some brewers to utilize novel ingredients. Theoretically the grains can be utilized as either malt or adjuncts, and the brewer's decision will at least be partially determined by beer style and sensory attributes. While malting obviously contributes to increased extract, FAN, color and enzymatic activity, a central question is whether it provides additional health benefits over the raw grain.

In summary, we know that ancient brewers utilized these grains, and recent research has demonstrated that quality malt and beer can be produced from ancient wheats. However, definitive statements on overall quality and certainly health promoting phytochemicals will involve the analysis of a much larger group samples representing multiple cultivars, subtypes and growing environments.

**Author Contributions:** A.F.: investigation, formal analysis and writing—original draft; S.S.: methodology and investigation, resources and supervision; and P.B.S.: conceptualization and writing—review and editing. All authors have read and agreed to the published version of the manuscript.

**Funding:** The current research received no external funding.

**Acknowledgments:** We thank Steve Zwinger for providing the wheat samples, John Barr for technical help with malting and malt analyses, Delane Olsen for assistance with grain testing, Kathy Christianson for DON analysis, Kristin Whitney for assistance with enzyme assays and AX analysis, Zhao Jin for helping with phenolic acid assays, and James Gillespie for his assistance with the LC/MS-Q-TOF.

**Conflicts of Interest:** The authors declare no conflict of interest.

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
