# Peer review of "Observations on the Malting of Ancient Wheats: Einkorn, Emmer and Spelt"

_fermentation, doi:10.3390/fermentation6040125_

Round 1

Reviewer 1 Report

A very good paper indeed, with a twist of describing the ancient species being re-invented for today's use together with good measure and mention of pros and cons. Some minor conceptual errors are evident as well as misspellings (check the PDF for details). However, the thing I personaly liked the most is the humble approach of the authors while making direct statements derived from their results with almost instant recognition of the need for broadening the research in specific points. There IS a lack of available high-tech research within scientific community considering these "old" wheats and other almost forgotten cultivated species humans used throughout the history, so I applaud to authors once more and to the fact they compounded with trending theme of craft brewing - pure synergy.

Best of luck,

Anonymous Barley Breeder

Author Response

pleas find attached

Reviewer 2 Report

Attached file
